# 4D Deep Learning for Multiple Sclerosis Lesion Activity Segmentation

**Nils Gessert**[1]                                    NILS.GESSERT@TUHH.DE
[1] *Institute of Medical Technology, Hamburg University of Technology, Germany*
**Marcel Bengs**[1]                                    MARCEL.BENGS@TUHH.DE
**Julia Krüger**[2]                              JULIA.KRUEGER@JUNG-DIAGNOSTICS.DE
[2] *jung diagnostics GmbH, Hamburg, Germany*
**Roland Opfer**[2]                            ROLAND.OPFER@JUNG-DIAGNOSTICS.DE
**Ann-Christin Ostwaldt**[2]          ANN-CHRISTIN.OSTWALDT@JUNG-DIAGNOSTICS.DE
**Praveena Manogaran**[3]                         PRAVEENA.MANOGARAN@USZ.CH
[3] *Department of Neurology, University Hospital Zurich and University of Zurich, Switzerland*
**Sven Schippling**[3]                                SVEN.SCHIPPLING@USZ.CH
**Alexander Schlaefer**[1]                              SCHLAEFER@TUHH.DE

## Abstract

Multiple sclerosis lesion activity segmentation is the task of detecting new and enlarging lesions that appeared between a baseline and a follow-up brain MRI scan. While deep learning methods for single-scan lesion segmentation are common, deep learning approaches for lesion activity have only been proposed recently. Here, a two-path architecture processes two 3D MRI volumes from two time points. In this work, we investigate whether extending this problem to full 4D deep learning using a history of MRI volumes and thus an extended baseline can improve performance. For this purpose, we design a recurrent multi-encoder-decoder architecture for processing 4D data. We find that adding more temporal information is beneficial and our proposed architecture outperforms previous approaches with a lesion-wise true positive rate of 0.84 at a lesion-wise false positive rate of 0.19.

**Keywords:** Multiple Sclerosis, Lesion Activity, Segmentation, 4D Deep Learning

## 1. Introduction

Multiple sclerosis (MS) is a chronic disease of the central nervous system where the insulating covers of nerve cells are damaged, often causing disability. Disease progression can be monitored in the brain using magnetic resonance imaging (MRI) with fluid attenuated inversion recovery (FLAIR) sequences (Rovira et al., 2015). Here, MS causes lesions which appear as high-intensity spots. Lesion activity, the appearance of new and enlarging lesions, is the most important biomarker for disease progression (Patti et al., 2015). Quantitative lesion parameters, such as volume and amount, require lesion segmentation which is still performed manually (García-Lorenzo et al., 2013) although it is time-consuming and associated with a high interobserver variability (Egger et al., 2017).

For conventional single-scan lesion segmentation, automated approaches using conventional (Roura et al., 2015) and deep learning methods have been proposed (Danelakis et al.,

2018). Lesion activity is derived from two scans from two different time points. Thus, one approach is derive lesion activity from individual segmentation maps. Since this approach is associated with large inconsistencies (García-Lorenzo et al., 2013), automated methods have used image differences (Ganiler et al., 2014) or deformation fields (Salem et al., 2018). Recently, a deep learning approach used a two-path 3D CNN for jointly processing the two volumes and predicting lesion activity maps (Krüger et al., 2019).

Processing two 3D volumes from two time points can be considered a 4D spatio-temporal learning problem. We hypothesize that extending the 4D context by adding a temporal history could improve lesion activity segmentation. MRI scans from the more distant past can be seen as an extended baseline providing additional information on the development of lesions and enable more consistent estimates. For this purpose, we design a new multi-encoder-decoder architecture using convolutional-recurrent units for temporal aggregation. We evaluate whether adding an additional time point from the past improves performance and compare our approach to models based on previous approaches.

## 2. Methods

**The dataset** we use is part of observational MS study at the University Hospital of Zurich, Switzerland. In total, we consider 44 MS cases where each case comes with a follow-up (FU), baseline (BL) and history (HS) FLAIR image. HS was acquired before BL. All scans are resampled to $1\,\mathrm{mm} \times 1\,\mathrm{mm} \times 1\,\mathrm{mm}$ and we rigidly register BL and HS to FU. Three independent raters labeled new and enlarging lesions using a tool showing both FU and BL. Thus, the task at hand is to predict a 3D lesion activity map $y \in \mathbb{R}^{H \times W \times D}$ which shows the lesion activity between BL and FU using a spatio-temporal tensor $x \in \mathbb{R}^{T \times H \times W \times D}$ consisting of FLAIR images. $H$, $W$ and $D$ are the spatial image dimensions and $T$ is the temporal dimension. Previous approaches used FU and BL ($T = 2$) while we investigate using FU, BL and HS ($T \geq 3$).

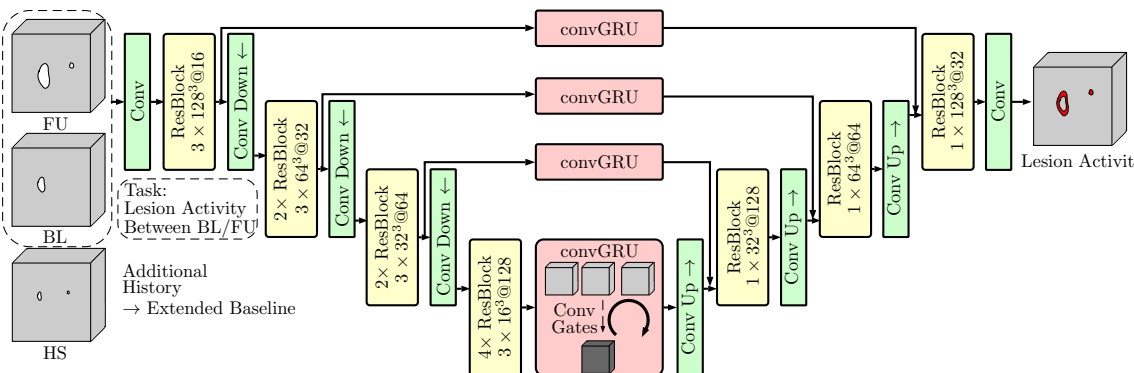

Figure 1: The deep learning model we propose. In each block we show the number of time points, spatial size and number of feature maps. The model receives FLAIR image volumes as the input.

The deep learning model we propose is a ResNet-based (He et al., 2016) multi-encoder-decoder 3D CNN architecture using convolutional gated recurrent units (convGRUs) for temporal aggregation, see Figure 1. The encoder processes all $T$ volumes individually and in parallel which can be interpreted as a $T$-path encoder where all paths share weights. Thus, the encoder path consists of 3D convolutional layers that process each volume in parallel. Then, all time points are aggregated using convGRU units. Finally, the decoder processes the aggregated 3D representation for predicting the lesion activity map. We also employ our convolutional-recurrent aggregation strategy for the long-range connections spanning from encoder to decoder. Note that there are previous recurrent models with similar naming to ours (Alom et al., 2019; Chen et al., 2016). Our problem, approach and model are fundamentally different as we use the recurrent units for temporal aggregation between encoder and decoder while previous methods used recurrence for spatial aggregation everywhere in the network. We compare our strategy to the previous approach of simply concatenating the volumes along the feature map dimension for processing in the decoder. The individual volume input size is $128 \times 128 \times 128$ with a batch size of 1. Due to the small batch size we use instance normalization (Ulyanov et al., 2016). Before training, we split off a validation set of 5 cases for hyperparameter tuning and a test set with 10 cases for evaluation. During training, we randomly crop subvolumes from the differently-sized scans. We train for 300 epochs using a learning rate of $\alpha = 10^{-4}$ and exponential learning rate decay. For evaluation, we use multiple, overlapping crops to form an entire lesion activity map for each case. Overlapping regions are averaged.

## 3. Results and Discussion

Table 1: Results for all experiments. We show the mean dice score, lesion-wise false positive rate (LFPR), false-positives (FPs) and lesion-wise true positive rate (LTPR). Lesions are defined as 27-connected components and any positive overlap between prediction and ground-truth is treated as a true positive.

| Model | Dice | LFPR | FPs | LTPR |
|---|---|---|---|---|
| Enc-Dec $T = 2$ (Krüger et al., 2019) | 0.62 | 0.30 | 1.1 | 0.81 |
| Enc-Dec $T = 3$ | 0.59 | 0.35 | 1.3 | 0.83 |
| Enc-convGRU-Dec $T = 2$ | 0.63 | 0.21 | 0.71 | 0.81 |
| Enc-convGRU-Dec $T = 3$ | **0.64** | **0.19** | **0.63** | **0.84** |

Our results are shown in Table 1. Our proposed model (Enc-convGRU-Dec) outperforms the previous approach (Enc-Dec) in terms of all metrics. Even for the conventional case with $T = 2$ (Krüger et al., 2019), our method improves performance. This suggests that recurrent aggregation might be preferable to time point concatenation. Using $T = 3$ instead of $T = 2$ also improves performance for our model which is not the case for Enc-Dec. Thus, when using a single additional scan, we already observe a slight performance improvement. This might indicate that an additional history is indeed beneficial and provides a more consistent baseline than a single scan. Furthermore, our architecture comes with the advantage that

arbitrary numbers of time points can be processed without changing the number of trainable model parameters. Therefore, future work could investigate the potential advantage of a longer history with a larger dataset containing more time points. Furthermore, our approach could be applied with other 4D segmentation problems.

## Acknowledgments

This work was partially supported by AiF grant number ZF4268403TS9 and ZF4026303TS9.

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
