# OpenReview forum: "4D Deep Learning for Multiple-Sclerosis Lesion Activity Segmentation"
_MIDL.io/2020/Conference — MIDL 2020_

### Official Review · AnonReviewer2 · 2020-03-03
**Incorporation of temporal data for improved segmentation**

**Rating:** 3
**Confidence:** 3

**Review:**

The paper proposes to include recurrent layers in an encoder-decoder architecture to improve the segmentation of lesion activity. The method is clearly justified and introduced and shows clear improvements over a baseline that concatenates encodings of different time points.

The extension of including recurrent connections for a temporal problems seems rather obvious but still requires effort to get working correctly. The simplicity of the approach, the applicability to different temporal applications, and convincing results make this paper a nice and interesting read.

The paper lacks clarity in the exact experimental setup and I personally would have liked a better introduction to lesion activity segmentation. Fig. 1 seems to suggest that the paper aims to segment the differences in lesion segmentations for two different time points. The figure might be slightly misleading assuming that HS is taken before HL. HS shows some lesion in the top right corner that isn't present in BL. Is that regular behaviour for those tasks?

- Furthermore, it seems that the authors did not use a validation set for developing and validating their method but might report results that are overfit to the test set.
- It is unclear how the full-volume activity maps are generated: how is are overlapping patches aggregated? Is there and consideration for boundary effects?
- The metrics seem not clearly defined: does FPs mean voxel-wise false positives? How are lesion-wise metrics defined?
- Lastly, do you ensure that the baseline has a similar capacity as the model including GRUs? Is there a similar memory footprint or number of model parameters? If not, this might not be a fair comparison.

It would be interesting to explore whether this model can generalise to unknown lengths of history. Also, could it be helpful to have intermediate segmentation or activity supervisions to use the varying time points as some training signal also?

---

### Official Review · AnonReviewer4 · 2020-03-04
**The paper proposes a GRU-based CNN which can leverage MRI scans from different time points for New and Enlarging Lesions.**

**Rating:** 3
**Confidence:** 3

**Review:**

The idea of the paper is good. Results support the idea and gives an increase in the performance compared to baseline methods. Considering the page limit, the paper is well written. It would be nice if authors can provide a reference for lesion-wise false positive rate (LFPR) and Lesion-wise true positive rate (LTPR).

---

### Official Review · AnonReviewer1 · 2020-03-06
**Identifying lesion change in MS**

**Rating:** 4
**Confidence:** 3

**Review:**

Authors present their work on identifying MS lesion change (appearance and enlarging lesions). This is a well-written abstract and an interesting method. The method uses GRU modules to include two or more images of the patient to identify lesion activity.

I assume the method processes FLAIR images, but this is not 100% clear to me. Figure 1 seems to suggest that lesion maps are fed into the model in stead of FLAIR images. Can authors clarify this?

Authors identify three time points for each subject: HS (an early scan), BL (baseline, comes after HS), and FU (follow-up, the most recent scan). In the results, models are compared on T=2 (BL and FU) and T=3 (BL, FU, and HS). T=3 seems to work better and authors suggest that the added history might help. However, an alternative hypothesis for this improved performance could be that the difference between HS and FU is much larger than BL and FU; hence adding HS works. It would be interesting to also add T=2 with HS and FU; because I suspect that it is just the longer time period between HS and FU that leads to increased lesion activity that is easier to detect.

It is unclear to me whether authors used a separate validation dataset for optimizing the hyperparameters of their model. Or that the reported results are on the test set that was also used to select the best performing parameters and results?

Did the human raters have access to HS when annotating lesion activity?

Authors look for 'new and enlarging' lesions (abstract): what about disappearing lesions?

---

### Official Review · AnonReviewer3 · 2020-03-12
**the method deals with a critical task while the technical novelty is somewhat limited**

**Rating:** 2
**Confidence:** 4

**Review:**

The authors proposed a 4D encoder-decoder CNN with convolutional recurrent gate units to learn multiple sclerosis (MS) lesion activity maps using 3D volumes from 2 time points. The proposed architecture connects the encoder and decoder with GRU to incorporate temporal information. It's compared to an earlier method which uses a 3D network and time-point concatenation and reports improvement in Dice scores, false positive rates and true positive rate.

The improvement gained by the proposed method validates the effectiveness of recurrent units, and the most significant gain is from the false positive rates.

Meanwhile, a few clarifications may be necessary:

1) in term of runtime, does the addition of GRUs take much more training time and memory comparing to the concatenation of 3D volumes?
2) what is the dimension of input, is it $H\times W \times D$ or $T\times H \times W \times D$ ?  If it's the latter one, is the convolution done with a 4D filter?
3) more details about the convGRU may be useful, for example its architecture.

Overall, the problem the paper tackles is critical, and the proposed network component is effective to some extent. The conclusion is more like a validation for the usefulness of the temporal information, while technical novelty may not be very sufficient in this case.

---

### Meta-Review · Area_Chair1 · 2020-04-05
**MetaReview of Paper92 by AreaChair1**

**Rating:** 3

**Metareview:**

The presented paper appears to be well written and presents interesting and promising results on an important problem. There is still room for improvement in the validation and clarification of the methods

**Paper Type:**

methodological development

---

### Decision · Program_Chairs · 2020-04-11

Accept